# Application of Oxidative Stress to a Tissue-Engineered Vascular Aging Model Induces Endothelial Cell Senescence and Activation

**DOI:** 10.3390/cells9051292

**Published:** 2020-05-22

**Authors:** Ellen E. Salmon, Jason J. Breithaupt, George A. Truskey

**Affiliations:** 1Department of Biomedical Engineering, Duke University, Durham, NC 27708, USA; ellen.salmon@duke.edu; 2Miller School of Medicine, University of Miami, Miami, FL 33136, USA; jjb239@miami.edu

**Keywords:** endothelial cells, vascular smooth muscle cells, senescence, oxidative stress, tissue-engineered blood vessel

## Abstract

Clinical studies have established a connection between oxidative stress, aging, and atherogenesis. These factors contribute to senescence and inflammation in the endothelium and significant reductions in endothelium-dependent vasoreactivity in aged patients. Tissue-engineered blood vessels (TEBVs) recapitulate the structure and function of arteries and arterioles in vitro. We developed a TEBV model for vascular senescence and examined the relative influence of endothelial cell and smooth muscle cell senescence on vasoreactivity. Senescence was induced in 2D endothelial cell cultures and TEBVs by exposure to 100 µM H_2_O_2_ for one week to model chronic oxidative stress. H_2_O_2_ treatment significantly increased senescence in endothelial cells and mural cells, human neonatal dermal fibroblasts (hNDFs), as measured by increased p21 levels and reduced NOS3 expression. Although H_2_O_2_ treatment induced senescence in both the endothelial cells (ECs) and hNDFs, the functional effects on the vasculature were endothelium specific. Expression of the leukocyte adhesion molecule vascular cell adhesion molecule 1 (VCAM-1) was increased in the ECs, and endothelium-dependent vasodilation decreased. Vasoconstriction and endothelium-independent vasodilation were preserved despite mural cell senescence. The results suggest that the functional effects of vascular cell senescence are dominated by the endothelium.

## 1. Introduction

Complications due to cardiovascular disease increase dramatically with age [1]. In particular, a decline in the endothelium-dependent vasoreactivity occurs, even in healthy, low-risk individuals, which predisposes them to cardiovascular disease [2]. As we age, vascular cells accumulate damage in a variety of ways. This damage results in cellular senescence, a phenotype in which cells have exhausted their proliferative capacity yet resist apoptosis [3,4]. For example, circulating lipoproteins damage and inflame the endothelium as well as accumulate within vascular smooth muscle cells. Periods of inflammation result in oxidative stress, reduced autophagy, and poor vasoreactivity [5]. 

Endothelial cells serve as key regulators of vessel tone and patency, blood flow, vessel permeability, immune response, and vascular smooth muscle cell (vSMC) quiescence [6]. Senescent endothelial cells exhibit lower levels of endothelial nitric oxide synthase (eNOS) activity [7]. The increased presence of reactive oxygen species (ROS) causes significant degradation of cytosolic nitric oxide (NO) [7,8,9]. The resulting limited NO bioavailability leads to reduced vasoreactivity and increased vSMC proliferation [10]. This contributes to clinically observed reductions in endothelium-dependent vasodilation in older individuals [11]. Additionally, endothelial cell senescence increases secretion of inflammatory signals, such as TNF-α, resulting in increased activation of the transcription factor nuclear factor kappa B (NF-kB) [12,13]. The resulting increase in the expression of VCAM-1, intracellular adhesion molecule 1 (ICAM-1), and other cellular adhesion markers facilitates the adhesion of circulating monocytes, directly contributing to the development of atherosclerosis. Furthermore, senescent endothelial cells have been identified inside atherosclerotic plaques [14]. 

vSMCs, which comprise the arterial wall, are also affected by cellular senescence. Senescence-associated β-Galactosidase and cyclin-dependent kinase inhibitor p21 positive vSMCs have been identified within the intima of advanced atherosclerotic plaques [15]. The presence of senescent vSMCs also increases the plaque area and the necrotic core within the vessel intima [16]. However, vSMC-dependent vasoreactivity is preserved in older patients [17]. H_2_O_2_ has been used in many studies to examine the effects of oxidative stress or stress-induced senescence on endothelial cells in vitro. In some studies, H_2_O_2_ is applied at a high dose, often in excess of 250 µM H_2_O_2_ for less than 12 h [18]. While this significantly increased the senescence of the endothelial cells, this dose of H_2_O_2_ is nearly eight times the highest estimated plasma concentrations of H_2_O_2_ [19]. Other studies have treated mouse brain microvascular endothelial cells with 50 µM H_2_O_2_, but a treatment period of 10 days was necessary for any significant increase in endothelial cell senescence [20].

There are numerous models to study cardiovascular disease ranging from *ApoE* knockout mice to microvascular endothelial networks and cell culture monolayers. While animal models are advantageous because they permit long-term studies and the use of clinically relevant measures to quantify outcomes, the utility of these systems to study senescence specifically is limited. In vitro models offer greater tunability, facilitating more precise applications of stressors to specific cell types rather than looking at systemic effects. Key aspects of endothelial cell behavior, such as nitric oxide synthesis and TFG-β production, are shear-mediated, limiting the utility of monolayer culture [21]. Notch and paracrine signaling between endothelial cells and vascular smooth muscle cells have potent effects on vessel behavior, necessitating a co-culture approach [22]. Tissue-engineered blood vessels (TEBVs) overcome these limitations by recreating a complete blood vessel in vitro within an optically clear perfusion system [23]. Their functionality has been well characterized and they are responsive to treatment with TNF-α and statins [24]. They provide an ideal foundation for the development of a three-dimensional in vitro vascular senescence model. In this study, we developed a TEBV model of vascular senescence and examined the relative influence of endothelial cell and smooth muscle cell senescence on vasoreactivity.

## 2. Materials and Methods

### 2.1. Cell Culture 

Human cord blood-derived endothelial colony forming cells (CBECFCs) were isolated as previously described [25]. Isolation and culture protocols for CBECFCs were approved by the Duke University Institutional Review Board. Human umbilical cord blood samples were obtained from the Carolina Cord Blood Bank. Patient identifiers were removed prior to receipt. 

After receipt, blood was diluted 1:1 with Hanks Buffered Salt Solution (HBSS, Invitrogen, Grand Island, NY, USA). The diluted blood sample was layered slowly atop an equal volume of room-temperature Histopaque 1077 (Sigma, Milwaukee, WI, USA) and centrifuged at 740× *g* for 30 min. Buffy coat mononuclear cells were removed and washed with Endothelial Cell Growth Medium (Cell Applications, St. Louis, MO, USA) containing 1% penicillin/streptomycin solution (Gibco, Waltham, MA, USA). The mononuclear cells were centrifuged at 515× *g* for 10 min and resuspended in endothelial cell culture media three times and then plated on TCPS six-well plates coated with 50 µg/mL rat tail collagen I. Media were changed daily for the first seven days of culture and every 48 h thereafter. Colony formation was visible after 7–10 days of culture. Flow cytometry was used to confirm the purity of the resulting endothelial cell population (Appendix A). Cells were harvested at passages 4–6. After resuspending cell samples in 10% goat serum, 5 μL of the appropriate antibody was added to a 95 μL sample containing 500,000 cells. Cells were then fixed with 0.5% PFA, rinsed, and resuspended in DPBS. In total, 9000 events per sample were recorded.

Human neonatal dermal fibroblasts (hNDFs) were purchased from Lonza (Walkersville, MA, USA). They were grown in high glucose Dulbecco’s modified Eagle medium (Invitrogen, Grand Island, NY, USA) supplemented with 10% (*v*/*v*) heat-inactivated fetal bovine serum (Gibco, Waltham, MA, USA), 1% penicillin/streptomycin (Gibco, Waltham, MA, USA), 1% MEM non-essential amino acids (Gibco, Waltham, MA, USA), 1% sodium pyruvate (Gibco, waltham, MA, USA), 1% GlutaMAX (Gibco, Waltham, MA, USA), and 0.1% β-Mercaptoethanol (ThermoFisher, Waltham, MA, USA). Media were changed every 48 h. 

### 2.2. Tissue-Engineered Blood Vessel (TEBV) Fabrication 

Tissue-engineered blood vessels (TEBVs) were made as previously described [24]. Rat tail collagen I (Corning, Bedford, MA, USA) was diluted to a concentration of 2.05 mg/mL in 0.6% acetic acid comprising 80% of the total volume of the TEBV solution. 10× serum-free Dulbecco’s Modified Eagle’s Medium (DMEM, Sigma, Milwaukee, WI, USA) was added to make up 10% of the total TEBV volume. The mixture was then titrated to a pH of 8.5 with 5 M NaOH. A solution of hNDFs at a concentration of 5 × 10^6^ cells/mL was added, comprising the final 10% of the TEBV volume, and leading to a final density of 0.5 × 10^6^ hNDFs/mL in the TEBV. After mixing, the gel solution was poured into a 3 mL luer-lok syringe (BD Biosciences, Franklin Lakes, NJ, USA) stoppered with a closed two-way stopcock. To create a tubular collagen scaffold, an 810 µm diameter steel mandrel was inserted into the center of the syringe mold and held in place with parafilm. The solution gelled at room temperature over 30 min. 

After gelation was complete, the gel and mandrel were removed from the syringe mold and laid on a 0.8 µm pore Nylon membrane filter (Whatman, Milwaukee, WI, USA) atop 10 autoclaved Kim Wipes. The TEBV rested there for 8–10 min until ~90% of the water was removed. The nylon filter and TEBV were then transferred to a 150 mm petri dish filled with sterile PBS to facilitate easier removal of the filter and lubricate the PBS for removal from the mandrel. Next, the TEBV was mounted in an acrylic chamber with hollow grips (0.711 µm outer diameter, McMaster-Carr, Elmhurst, IL, USA) on each end of the vessel to allow perfusion of the vessel lumen. Black silk sutures held the TEBV in place on the grips (Appendix A). Next, a solution of 500,000 ECFCs suspended in endothelial cell growth media was slowly pushed through the vessel lumen. The vessel was rotated vertically for 60 min at a speed of 10 rph to ensure uniform monolayer adhesion. TEBVs were perfused with TEBV growth media at a flow rate of 0.5 mL/min for the first 24 h after fabrication, after which the flow rate was increased to 2 mL/min for the remainder of the experiment (Appendix A). This resulted in a laminar shear stress of 6.8 dynes/cm^2^ [24]. The TEBV perfusion medium contained low glucose DMEM, 1% penicillin/streptomycin, 1% MEM non-essential amino acids, 0.1% β-Mercaptoethanol, and 3.7% heat-inactivated fetal bovine serum for the first week of culture. After one week of culture, 2 mg/mL ε -aminocaproic acid was added to preserve the long-term integrity of the TEBV.

### 2.3. Vasoreactivity Testing of TEBVs

TEBV vasoreactivity was quantified in response to phenylephrine, acetylcholine, and sodium nitroprusside. Stock solutions (1 mM) of each drug were prepared by diluting lyophilized powders in sterile DPBS without calcium or magnesium. This 1 mM solution was injected into the reservoir of the TEBV flow circuit via a silicone injection port integrated into the perfusion tubing. A 1 µL/mL dilution was used, resulting in a dose of 1 µM in the flow circuit. These doses were expected to elicit about half the maximum vasodilation or vasoconstriction of the TEBVs. The TEBV diameter equilibrated after the introduction of phenylephrine for five minutes. Next, acetylcholine was injected, and the diameter change was quantified after five minutes. Finally, sodium nitroprusside was added to the flow circuit, and the diameter change was quantified after 8 min. Diameter changes were recorded on video with a stereo microscope equipped with a 0.5× magnification lens using ISCapture. Still shots were selected from these videos at the time points described above (300 s after phenylephrine injection, 300 s after acetylcholine injection, and 480 s after sodium nitroprusside injection). The diameter at the time of phenylephrine injection was used as a baseline. The still images were quantified using the line tool in ImageJ. Five points evenly spaced along the TEBV were randomly selected to measure the diameter. The same points along the TEBV were used to measure the diameter change in response to each drug. A typical TEBV has a baseline diameter of 1500–2000 μm, corresponding to about 150–250 pixels in ImageJ.

TEBV vasoreactivity was first quantified seven days after fabrication to evaluate baseline vasoreactivity and to confirm the appropriate function of the TEBV. For this initial test of vasoreactivity, only phenylephrine and acetylcholine were used. If vasoconstriction and vasodilation did not exceed −1% and 1%, respectively, the vessel was not used for further study. In the data presented here, this test was called “Day 0”, and further tests of vasoreactivity were described using the number of days after this initial test of vasoreactivity, rather than the total number of days since vessel fabrication. H_2_O_2_ was added after a vessel passed this initial test of vasoreactivity on Day 7, and vasoreactivity was measured again on Day 5 or Day 7 of H_2_O_2_ treatment with acetylcholine, phenylephrine, and sodium nitroprusside.

### 2.4. Hydrogen Peroxide Treatment

ECFCs or hNDFs were plated at a density of 5300 (controls) or 7900 cells/cm^2^ (H_2_O_2_ treated) in 24-well plates. The lower seeding density of the control cells was necessary to reduce over-crowding of the cells within the wells due to the higher growth rate of the cells in the absence of H_2_O_2_. All cells were allowed 18–24 h to firmly adhere to the plate before H_2_O_2_ was introduced to the cell culture media. H_2_O_2_ was diluted from a 30% stock solution (9.77 M) to 50 or 100 µM in the appropriate cell culture growth medium. Media were changed every 48 h until the cells were fixed on the seventh day of treatment. TNF-α was diluted from a 200 U/µL stock to 100 U/mL in the appropriate cell culture and added with media containing H_2_O_2_ for the final 24 h of cell culture.

The degradation of H_2_O_2_ in the media was estimated using a fluorometric hydrogen peroxide assay kit (Sigma MAK165, St. Louis, MO, USA). H_2_O_2_ stocks were made according to assay protocols and rested in the cell culture incubator for 24 or 48 h before analysis. At the appropriate time point, samples were diluted along a standard curve ranging from 0.1–10 µM H_2_O_2_ in assay buffer. Fluorescence at 590 nm was measured using a Perkin–Elmer Victor^3^ 1420 Multilabel Counter. 24- and 48-h samples were compared to freshly prepared H_2_O_2_ stocks (Appendix A).

### 2.5. Immunofluorescence

Cells in well plates were fixed with 10% formalin for 10 min at room temperature. Cells were permeabilized with 0.1% Triton-X and then blocked with 10% goat serum (Gibco, Waltham, MA, USA) in Dulbecco’s Phosphate Buffered saline (DPBS, Sigma, Milwaukee, WI, USA). Primary antibodies for P21 and VCAM-1 (Abcam, Boston, MA, USA) were diluted 1:250 in 10% goat serum. Primary antibodies for E-Selectin and ICAM-1 (SCBT, Dallas, TX, USA) were diluted 1:200 in 10% goat serum. After staining overnight at 4 °C, samples were rinsed three times with DPBS. Goat anti-rabbit Alexa-Fluor 594 or goat anti-mouse Alexa-Fluor 488 secondary antibodies were added at a dilution of 1:500 in 1% Goat serum with 1 µL/mL Hoescht 33342 for 1 h at room temperature. Samples were rinsed an additional three times in DPBS and imaged immediately. 2-D samples were imaged at 20× on a Nikon Eclipse TE2000-U. TEBV sections were mounted on slides with Fluor Save, covered with a cover slip, and imaged on a Zeiss 510 inverted confocal microscope at 20× magnification.

For TUNEL staining, TEBVs were fixed on Day 14 (after seven days of 0 or 100 μM H_2_O_2_) in 10% formalin and cut into 1 cm sections. Sections were embedded in O.C.T. (Tissue-Tek, Torrance, CA, USA) and frozen at −80 °C. Sections (10 µm) were cut using a Leica CM 1950 and were mounted on Superfrost Plus slides (VWR, Aurora, OH, USA). After mounting, sections were permeabilized with Proteinase k and stained using an Alexa-Fluor 594 TUNEL kit (ThermoFisher, Waltham, MA, USA). Then, 30-min DNase I treatment (ThermoFisher, Waltham, MA, USA) was administered to sections after permeabilization to provide a positive control. After TUNEL staining, sections were blocked with 3% BSA in 1X PBS. Sections were then stained for α-SMA in 10% goat serum at 4 °C overnight and labeled with Alexa Fluor 488 secondary antibodies. A 1× solution of Hoescht 33342 was used to stain nuclei. Three sections from each vessel were imaged for analysis.

### 2.6. qRT-PCR

RNA was extracted from the TEBVs using an RNeasy Mini Kit (Qiagen, Germantown, MD, USA) and a slightly modified version of the manufacturer’s protocol. Briefly, TEBVs were submerged in 150 µL Buffer RLT with 10 µL/mL β-mercaptoethanol and vortexed for 2 min to disrupt tissue. Then, 295 µL RNase/DNase-free water and 15 µL proteinase K (Qiagen, Germantown, MD, USA) were added, and the solution was incubated at 55 °C for 10 min. The solution was centrifuged at 10,000× *g* for three minutes, and the supernatant was used for further extraction. Subsequently, 225 µL of RNase/DNase free 98% ethanol was added, and 700 µL of this solution was added to an RNeasy spin column. This was centrifuged at 10,000× *g* for 15 s. The column was washed with the remaining RNeasy mini kit buffers as per the manufacturer’s protocol and was eluted in 40 µL of RNase/DNase free water. RNA purity and concentration were assessed using a NanoDrop spectrophotometer. Reverse transcription of RNA into cDNA was performed with 250 ng of TEBV RNA using the iScript cDNA Synthesis Kit (BioRad, Hercules, CA, USA). RT-PCR was performed using iQ SYBR Green Supermix (Bio-Rad, Hercules, CA, USA) and the CFX96 Connect Real-Time PCR Detection System (Bio-Rad, Hercules, CA, USA). Housekeeping gene primers were glyceraldehyde 3-phosphate dehydrogenase (GAPDH) primers (VHPS-3541, RealTime Primers, Elkins Park, PA, USA). Primers for the gene of interest were custom made (Integrated DNA Technologies, Coralville, ID, USA). Sequences used were endothelial nitric oxide synthase (NOS3) Fwd: 5′-CATCTTCAGCCCCAAACGGA-3′ and Rev: 5′-ACGGGATTGTAGCCTGGAAC-3′, inducible nitric oxide synthase (NOS2) Fwd: 5′-CCCCCAGCCTCAAGTCTTATTTC-3′ and Rev: 5′-CAGCAGCAAGTTCCATCTTTCAC-3′, Nf-κB p65 (v-rel avian reticuloendotheliosis viral oncogene homolog A—RELA) Fwd: 5′-AGCTCAAGATCTGCCGAGTG-3′ and Rev: 5′-ACATCAGCTTGCGAAAAGGA-3′, Sirtuin 1 (SIRT1) Fwd: 5′-TGCTGGCCTAATAGAGTGGCA-3′ and Rev: 5′-CTCAGCGCCATGGAAAATGT-3′, NOX4 Fwd: 5′- GCAGGATCCGTCATAAGTCATCCCTCAGA-3′ and Rev: 5′-GCTGTTAACGTCGACTCAGCTGAAAGACTCTTTAT-3′, von Willebrand factor (vWF) Fwd: 5′-GCAGTGGAGAACAGTGGTG-3′ and Rev: 5′-GTGGCAGCGGGCAAAC-3′, prolyl-4-hydroxylase β (P4HB) Fwd: 5′-GGACGTGGAGTCGGACTCTG-3′ and Rev: 5′-GGCTGTCTGCTCGGTGAACT-3′, fibroblast specific protein-1 (FSP-1) Fwd: 5′-GATGAGCAACTTGGACAGCAA-3′ and Rev: 5′-CTGGGCTGCTTATCTGGGAAG-3′. The fold change from reference RNA was calculated as previously described [26].

### 2.7. Statistical Analysis

Statistical analyses were performed using JMP Pro 15 (SAS Institute). One- or two-way ANOVA with post-hoc Dunnett’s test was used to compare means for all immunofluorescence quantification. A repeated measures ANOVA with post-hoc Tukey’s test was used to compare means for all TEBV vasoreactivity data. P values less than 0.05 were considered significant. All data shown graphically represent the mean ± SEM. N represents the number of independent experiments.

## 3. Results

### 3.1. H_2_O_2_ Treatment Caused Senescence and Inflammation in Endothelial Colony-Forming Cells (ECFCs)

Senescence in CBECFCs was measured by immunofluorescence of the cell-cycle inhibitor p21 [27] and quantification of p21 positive nuclei as a percentage of total nuclei. Five-day treatment with either 50 or 100 µM H_2_O_2_ did not affect the percentage of senescent cells (Figure 1A). Treatment with 100 µM H_2_O_2_ for seven days caused a significant increase in the percentage of senescent CBECFCs (Figure 1A,B). Treatment for five days, rather than seven days, or use of 50 μM H_2_O_2_ rather than 100 μM H_2_O_2_ failed to cause an increase in the percentage of p21-positive nuclei. A small percentage of CBECFCs was P21-positive on Day 5 and Day 7 even without application of H_2_O_2_. P21 expression was not measured on Day 0, but we believe this reflects the baseline senescence for these cell lines. There was no significant increase in senescence between Day 5 and Day 7 in the control cells.

H_2_O_2_ treatment at any concentration had no effect on the expression of ICAM-1 (Figure 2A,D). Introduction of 100 U/mL TNF-α significantly increased ICAM-1 expression, but H_2_O_2_ did not affect ICAM-1 levels alone or in combination with TNF-α (Figure 2A). VCAM-1 expression was minimally affected by treatment with 100 U/mL TNF-α, although there was a significant interaction effect between TNF-α treatment and 100 µM H_2_O_2_ (Figure 2B,D). Five-day treatment with H_2_O_2_ did not significantly affect VCAM-1 expression unless co-treated with 100 U/mL TNF-α. Seven-day treatment with 100 µM H_2_O_2_ caused a significant increase in VCAM-1 expression (Figure 2B). Treatment with 100 µM H_2_O_2_ for seven days caused a significant increase in E-Selectin expression that was comparable to TNF-α induced E-Selectin expression in controls (Figure 2C,D). Treatment with 100 U/mL TNF-α also caused significant E-selectin expression in ECFCs cultured for seven days, regardless of H_2_O_2_ concentration (Figure 2C,D). Of note, E-selectin expression in cells treated for five days with 100 µM H_2_O_2_ and given 100 U/mL TNF-α was higher than that in cells treated with other concentrations of H_2_O_2_ for five days (Figure 2C). A higher percentage of cells expressed VCAM-1 than E-Selectin.

### 3.2. Effects of H_2_O_2_ Exposure on hNDFs

Nuclear expression of p21 in hNDFs treated with 100 µM H_2_O_2_ for seven days was increased (Figure 3A). Expression of calponin and α-SMA appear unaffected by treatment with 100 µM H_2_O_2_ (Figure 3A). Quantification of these immunofluorescent images confirms that, while the senescence of the hNDF population increased considerably, the percentage of cells expressing the contractile proteins calponin and α-SMA was unaffected (Figure 3B). Treatment with 50 μM H_2_O_2_ and 5-day treatment durations were not tested.

### 3.3. Endothelium-Dependent Vasoreactivity Was Compromised in TEBVs Treated with H_2_O_2_

As described in the methods, all TEBVs were matured in normal growth media for seven days with a flow rate that produced a shear stress of 6.8 dynes/cm^2^ on the endothelium. As described in the methods, vasoreactivity was tested after this initial maturation and denoted as “Day 0”. Then, TEBVs were treated with 0 or 100 μM H_2_O_2_ for five or seven days. The 100 μM H_2_O_2_ treatment did not affect vasoconstriction in response to 1 μM phenylephrine compared to controls (Figure 4A). In contrast, TEBVs treated with 100 µM H_2_O_2_ for five days exhibited a significant reduction in endothelium-dependent vasodilation compared to the 0 µM H_2_O_2_ controls (Figure 4B). To assess whether the loss of endothelium-dependent vasodilation was the result of endothelial dysfunction specifically, on Day 12 or Day 14, TEBVs were also dosed with 1 µM sodium nitroprusside. In vessels treated with 100 µM H_2_O_2_ for five days there was a corresponding increase in their vasodilation in response to sodium nitroprusside. TEBVs matured in control media dilated in response to 1 μM acetylcholine but exhibited no change in their diameter in response to the subsequent introduction of sodium nitroprusside. This suggests that the maximum dilatory capacity of the vessel for these drug doses was reached. The endothelium-independent vasodilation caused by sodium nitroprusside in H_2_O_2_-treated TEBVs was not significantly different from the endothelium-dependent vasodilation observed in controls. This indicates that treatment with 100 µM H_2_O_2_ only caused significant dysfunction in the CBECFCs, but not in the hNDFs. When TEBVs were treated with 100 µM H_2_O_2_ for a full seven days, endothelium-dependent vasodilation was further reduced, and vasoconstriction in response to acetylcholine was reduced in several cases (Figure 4D). This was to be expected, as acetylcholine triggered constriction in the hNDFs in the absence of endothelial cells (Appendix A). Conversely, endothelium-dependent vasodilation improved slightly in vessels matured in control media. The endothelium-independent dilation of TEBVs treated with 100 µM H_2_O_2_ for seven days was comparable to the endothelium-dependent dilation observed in TEBVs matured in only 0 µM H_2_O_2_ (Figure 4D). Vasoconstriction in response to phenylephrine remained consistent (Figure 4C).

### 3.4. H_2_O_2_ Treatment Caused Endothelial Cell Senescence and Inflammation in TEBVs

Analysis of immunofluorescent-stained TEBV segments cut en face was used to confirm that the endothelium was both present and senescent, and that contractile protein expression in the hNDFs was maintained. Immunostaining showed that expression of p21 was significantly increased in the hNDFs encapsulated within the vessel wall after seven days of treatment with 100 µM H_2_O_2_ (Figure 5A,B). Expression of calponin and α-SMA in the vessel wall was not significantly affected (Figure 5A,B). H_2_O_2_ exposure also significantly increased the senescence of the ECFCs lining the vessel wall (Figure 5A,C). The expression of VCAM-1 in the endothelium was also significantly increased in vessels treated with 100 µM H_2_O_2_ compared to 0 µM controls (Figure 5A,C). It is worth noting that the number of ECFCs identified (e.g., vWF positive area coverage) had no observable differences after 7-day treatment with 100 µM H_2_O_2_. Furthermore, cross-sectional TUNEL staining confirmed that treatment with 100 µM H_2_O_2_ for seven days did not affect the proportion of apoptotic cells in the TEBVs (Figure 6). These data further confirm that the observed changes in vasoreactivity after 100 µM H_2_O_2_ treatment were the result of endothelial cell senescence, not the removal or death of the seeded endothelium.

Next we examined gene expression in TEBVs on Day 14 of maturation and compared those that had been treated with 100 μM H_2_O_2_ for seven days to those matured only in control media. Relative to controls, the 100 µM H_2_O_2_ treatment decreased the mRNA expression of NOS3 (1.01 ± 0.05 vs. 0.43 ± 0.08, *p* ≤ 0.01), the gene which codes for endothelial nitric oxide synthase (eNOS). Conversely, expression of NOS2, the gene that codes for inducible nitric oxide synthase (iNOS), was increased (1.01 ± 0.05 vs. 2.81 ± 0.55, *p* = 0.01) (Figure 7A). Treatment had no effect on the mRNA expression of RELA (1.02 ± 0.06 vs. 0.87 ± 0.05, *p* = 0.10), indicating no difference in the expression of the p65 subunit of nuclear factor-κB (Nf-κB). Interestingly, SIRT1 mRNA, which produces the anti-aging sirtuin 1, was more than doubled (1.01 ± 0.06 vs. 2.35 ± 0.25, *p* ≤ 0.01) after H_2_O_2_ treatment (Figure 7A). NOX4, the gene that codes for ROS-producing NADPH oxidase-4, was decreased by exogenous H_2_O_2_ (1.01 ± 0.05 vs. 0.18, *p* ≤ 0.01).

The relative expression of CBECFC-specific genes (vWF) was decreased in TEBVs treated with 100 μM H_2_O_2_ for seven days (Figure 7B). This difference in gene expression seemed not to have a significant effect on the amount of vWF in H_2_O_2_-treated vessels based on immunostaining (Figure 5A). There was also a significant increase in the expression of the hNDF-specific genes P4HB and FSP-1 (Figure 7B).

## 4. Discussion

We developed a TEBV model of vascular senescence and showed that treatment of TEBVs with 100 µM H_2_O_2_ for five to seven days diminished endothelium-dependent vasodilation while preserving endothelium-independent vasoreactivity (Figure 4), consistent with findings in vivo [11]. This study is the first to develop a tissue-engineered blood vessel model for senescence. Furthermore, this model can be analyzed using a non-destructive, clinically relevant metric of vascular health. We explored the relative contributions of endothelial cells and vascular smooth muscle cells to overall vasoreactivity. H_2_O_2_ (100 μM) was used to induce senescence in the TEBVs. This dose exceeds physiological plasma H_2_O_2_ concentrations [19]. This slightly higher concentration of H_2_O_2_ allowed us to accelerate the timeline along which senescence is induced without causing apoptosis or acute cellular damage.

In both 2D cell culture and the TEBVs, 100 µM H_2_O_2_ treatment for seven days caused significant senescence in CBECFCs and hNDFs (Figure 1). Although the H_2_O_2_ treatment did cause a significant increase in hNDF senescence, the expression of calponin and α-SMA was unaffected (Figure 3). When 100 μM H_2_O_2_ was applied to the TEBVs, endothelium-dependent vasoreactivity was significantly reduced, while endothelium-independent vasoreactivity was preserved (Figure 4). The robust sodium nitroprusside response confirmed that the dilatory capacity of the vessels had not been reduced, indicating that the observed loss of dilation is based on the effects of the endothelium specifically, rather than the hNDFs. Interestingly, TEBVs experienced a significant reduction in endothelium-dependent vasoreactivity after 5-day exposure to 100 µM H_2_O_2_, even though this was not long enough to produce a significant increase in endothelial cell senescence in 2D cultures. TEBV sections were stained for p21 and vWF, confirming that 7-day treatment with 100 μM H_2_O_2_ caused a significant increase in the number of senescent ECs, as well as hNDFs. Additionally, expression of eNOS was decreased in treated TEBVs. This suggests that the loss of endothelium-dependent vasodilation is the result of endothelial cell senescence rather than effacement of the endothelium. TUNEL staining also confirmed that 7-day treatment with H_2_O_2_ did not induce apoptosis in either the hNDFs or the ECs (Figure 6). Expression of calponin and α-SMA was unaffected by H_2_O_2_ treatment in the TEBVs, just as seen in 2-D cultures. This is consistent with the robust hNDF-dependent vasoreactivity of the TEBVs regardless of H_2_O_2_ concentration.

The endothelium also expressed the adhesion molecules VCAM-1 and E-Selectin after treatment with H_2_O_2_, confirming that this model also captures the pro-inflammatory effects of senescence on the vasculature (Figure 6). Of note, ICAM-1 expression was not increased by H_2_O_2_ treatment in 2D. Overall, the level of stress-induced E-selectin expression, though significantly higher than that in the controls, was less than the amount of stress-induced VCAM-1 expression. When HUVECs were treated with 50 U/mL TNF-α or IL-1, E-selectin expression peaked 4–6 h after treatment and remained only slightly above baseline after 48 h [28,29]. Conversely, ICAM-1 and VCAM-1 expression increased during 24-h exposure and remained high after 48 h [28]. This explains the lower expression of E-selectin compared to VCAM-1 and ICAM-1 in the TNF-α-treated positive controls (Figure 2), since these were measured 24 h after initiation of treatment. The absence of oxidative stress-induced ICAM-1 expression in the CBECFCs suggests it was not activated by H_2_O_2_ treatment; in other words, the expression was not low due to transient activation. ICAM-1 and VCAM-1 are both targets of NF-κB, the primary signaling molecule governing endothelial cell activation [30]. However, expression of the two adhesion molecules can be differentially regulated despite their shared upstream effector [28].

VCAM-1 expression in the TEBV endothelium was significantly increased after treatment with 100 µM H_2_O_2_ (Figure 5). Interestingly, RELA mRNA was not increased as would be suggestive of an inflammatory phenotype. However, no increase in mRNA coding for the p65 subunit of Nf-κB is not definitive proof that inflammation did not increase. Post-translational processing and modulation of Nf-κB activity were not measured in this study. Another interesting finding is that SIRT1 expression increased in H_2_O_2_-treated TEBVs. Sirtuin 1 is associated with DNA repair and cellular health. In HUVECs, H_2_O_2_ treatment decreased SIRT 1 expression [31]. It is unclear whether the bulk of SIRT1 mRNA in the present model was from hNDFs or ECFCs.

Treatment with 100 µM H_2_O_2_ did not affect vasoconstriction in response to 1 μM phenylephrine in the TEBVs (Figure 3A,C). In contrast, human saphenous vein samples showed a reduction in vasoconstriction ex vivo when treated with as little as 10 μM H_2_O_2_ for 16 h [32]. It is important to note that these data were obtained using a much higher concentration of noradrenaline (100 μM) as a vasoconstrictor, rather than the low dose of phenylephrine (1 μM) used here. It is possible that application of a higher dose of phenylephrine to the TEBVs could reveal a slight difference in the vasoconstriction with 100 μM H_2_O_2_ treatment. Furthermore, the saphenous vein is comprised of both vSMCs and perivascular fibroblasts, while the TEBVs used here contained only hNDFs within the vessel wall [33]. The presence of the vSMCs may add greater oxidative stress sensitivity to the endothelium-independent measures of vasoreactivity.

Several cell types could be used for the mural cells of TEBVs (vSMCs, mesenchymal stem cells (MSCs, and hNDFs), as well as for the endothelium (HAECs or ECFCs). While primary vSMCs seem ideal to maximize physiological relevance, they have limited proliferative capacity after isolation. Donor availability is also limited, and the quality of cells varies significantly from donor to donor [34]. If vSMCs from elderly donors are used, cell quality deteriorates even further [35]. MSCs are a more attractive cell type since they are more proliferative than vSMCs and differentiate to express vSMC proteins. However, TEBVs fabricated with hNDFs are significantly more contractile than those fabricated from MSCs [24] and still express the key proteins calponin and αSMA. While HAECs are readily available commercially, CBECFCs proliferate more quickly. These two cells are also functionally equivalent as characterized by the expression of nitric oxide, cellular adhesion in co-cultures, and shear-responsiveness [36]. CBECFCs are easily obtained with high yields per donor, and donor-to-donor variability is minimal, making them an ideal source of primary ECs for use in the TEBVs. The TEBVs were made with hNDFs and CBECFCs to maximize the functional capacity of the TEBVs in their “healthy” state. This allows the greatest sensitivity when analyzing vasoreactivity for loss and reduction of function.

Endothelial cells play a significant role in the maintenance of vascular health. As the endothelium grows dysfunctional, the risk of developing atherosclerosis increases significantly. The role of endothelial cell senescence in promoting an atherogenic phenotype in the vasculature is recognized. An optimal in vitro model of vascular aging will facilitate the development of novel treatments, such as senolytics, that specifically target the effects of cellular aging on vascular health. This study demonstrates the utility of tissue-engineered blood vessels (TEBVs) in generating an atherogenic environment. In future studies, the atherogenic potential of this model will be further explored by introducing monocytes and lipids to the TEBV perfusion circuit. H_2_O_2_-treated TEBVs will also be used to investigate the effects of drugs specifically targeting senescence (i.e., senolytics) on limiting atherogenesis.

## Figures and Tables

**Figure 1 cells-09-01292-f001:**
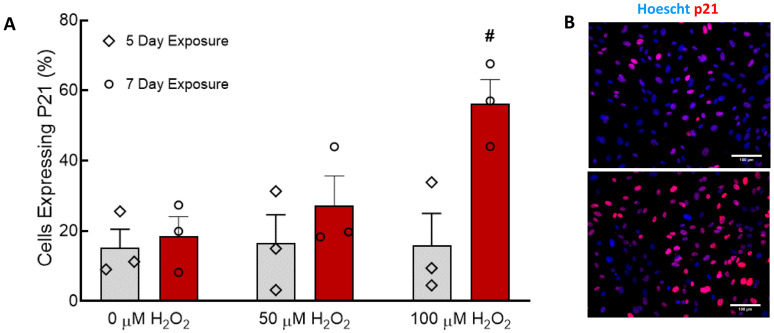
Treatment of cord-blood derived endothelial colony forming cells (CBECFCs) with 100 µM H_2_O_2_ for seven days induced CBECFC senescence. (**A**) The number of senescent (p21 positive) ECFCs was quantified after treatment for five or seven days with H_2_O_2_. H_2_O_2_ concentration had a significant effect on p21 expression (*p* < 0.05). Treatment with 100 µM H_2_O_2_ for seven days caused a significant increase in senescence compared to the 0 µM control (# *p* < 0.01). Bar graphs represent the mean ± SEM, individual points overlaid. N = 3–4. (**B**) Control ECFCs exhibited minimal p21 expression, while ECFCs treated with 100 µM H_2_O_2_ for seven days had high levels of nuclear p21 staining. Scale bars represent 100 µm.

**Figure 2 cells-09-01292-f002:**
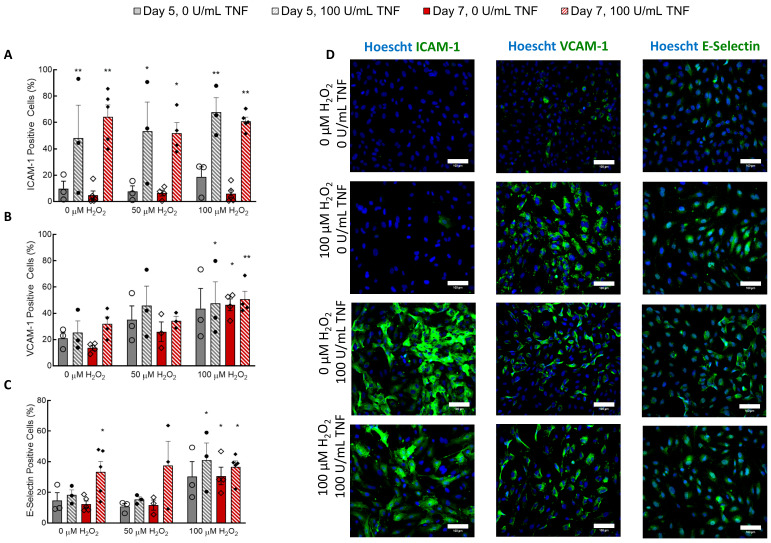
H_2_O_2_ treatment induced VCAM-1 and E-selectin expression in ECFCs. (**A**) TNF-α increased ICAM-1 expression (*p* < 0.0001), but H_2_O_2_ treatment did not affect ICAM-1 expression. (**B**) Both TNF-α and H_2_O_2_ affected VCAM-1 expression (*p* < 0.05). Post-hoc analysis shows that 7-day treatment with 100 µM H_2_O_2_ with or without TNF-α caused a significant increase in VCAM-1 expression. Five-day treatment with H_2_O_2_ caused a significant increase in VCAM-1 expression when combined with a 24-h dose of TNF-α. (**C**) Both TNF-α and H_2_O_2_ affected E-Selectin expression (*p* < 0.05). Seven-day treatment with 100 µM H_2_O_2_ caused a significant increase in E-Selectin expression with or without TNF-α treatment. (**D**) Immunofluorescence was used to obtain the results presented in A–C. Images show cells imaged on Day 7. Scale bars represent 100 µm. * *p* < 0.05 compared to 0 µM H_2_O_2_ and 0 U/mL TNF-α for each day. ** *p* < 0.005 compared to Day 5 0 µM H_2_O_2_ and 0 U/mL TNF-α. N = 3–5. Bars represents mean ± SEM.

**Figure 3 cells-09-01292-f003:**
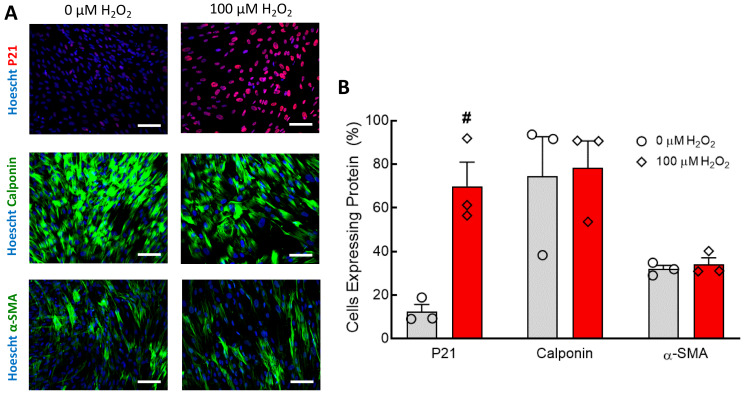
Seven-day 100 µM H_2_O_2_ treatment increased senescence in hNDFs but did not affect contractile protein expression. (**A**) hNDFs were stained for P21, Calponin, and α-SMA. P21 expression was higher in hNDFs treated with 100 µM H_2_O_2_. Scale bars represent 100 µm. (**B**) Quantification of p21, calponin, and α-SMA immunofluorescent staining. Seven-day treatment with 100 µM H2O2 caused a significant increase in hNDF senescence compared to 0 µM controls (# *p* < 0.001). Expression of calponin and α-SMA was not affected by 100 µM H2O2. N = 3. Bars represent mean ± SEM.

**Figure 4 cells-09-01292-f004:**
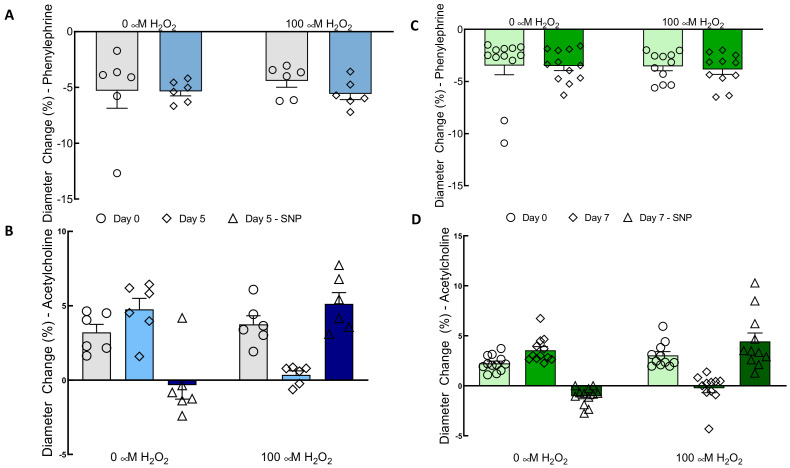
Treatment with 100 µM H_2_O_2_ for five or seven days eliminated endothelium-dependent vasodilation without affecting endothelium-independent vasoreactivity in TEBVs. For all conditions, TEBVs were perfused for seven days following fabrication before the start of the experiment with or without 100 µM H_2_O_2_ and perfusion. Day 0 thus represents seven days of perfusion. (**A**) Vasoconstriction in response to 1 μM phenylephrine was not affected by H_2_O_2_. (**B**) Vasodilation in response to 1 µM acetylcholine was reduced by H_2_O_2_ (*p* < 0.01), and the effect of time (i.e., date tested) was H_2_O_2_ dependent (*p* < 0.001). There was a significant difference between the acetylcholine response of 0 and 100 µM H_2_O_2_ TEBVs (*p* < 0.005). Sodium nitroprusside (SNP) mediated vasodilation in vessels treated with 100 µM H_2_O_2_ was not significantly different from acetylcholine-mediated vasodilation in vessels matured in 0 µM H_2_O_2_. (**C**) Vasoconstriction in response to phenylephrine was not affected by H_2_O_2_ concentration or the timepoint, even when the H_2_O_2_ treatment was extended from five to seven days. (**D**) Vasodilation in response to acetylcholine was reduced by H_2_O_2_ (*p* < 0.0001). Vessels treated with 100 µM H_2_O_2_ from Day 0 to Day 7 constricted in response to acetylcholine, rather than dilated. Sodium nitroprusside-mediated vasodilation in TEBVs matured in 100 µM H_2_O_2_ was not significantly different from acetylcholine-mediated vasodilation in control vessels. N = 7 (**A**,**B**). N = 12 (**C**,**D**). Bars represent the mean ± SEM.

**Figure 5 cells-09-01292-f005:**
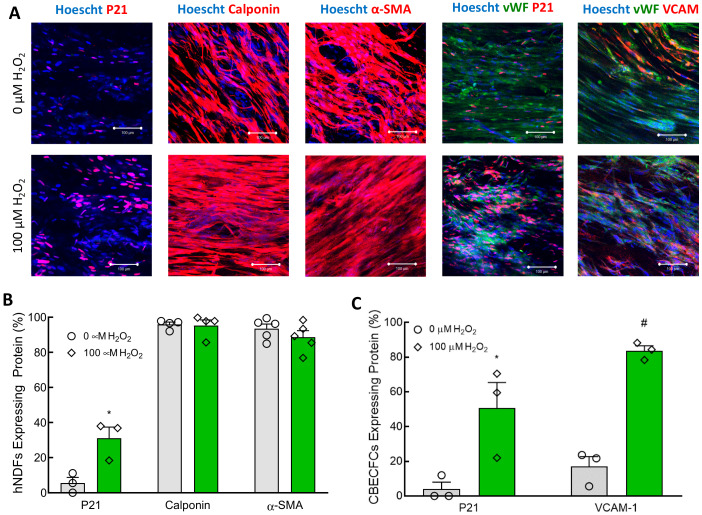
Seven-day treatment of TEBVs with 100 µM H_2_O_2_ caused senescence in the embedded hNDFs and ECFCs, as well as an increase in endothelial VCAM-1 expression. The treatment regime was the same as that described in Figure 4. (**A**) Immunofluorescence was used to evaluate the expression of p21, calponin, α-SMA, and VCAM in the appropriate cell types within the TEBVs. Expression of p21 and VCAM-1 was increased in TEBVs treated with 100 µM H_2_O_2_ for one week. Scale bars represent 100 µm. (**B**) Quantification of immunofluorescent images confirmed that hNDFs encapsulated in TEBVs experienced a significant increase in p21 expression (* *p* < 0.05) with 100 µM H_2_O_2_ treatment, but expressions of calponin and α-SMA were not affected. Values represent the mean ± SEM. N = 5. (**C**) Quantification of immunofluorescent images confirmed that treatment of TEBVs with 100 µM H_2_O_2_ significantly increased the expressions of p21 and VCAM-1 in the CBECFC monolayer (# *p* < 0.005). Bars represent the mean ± SEM. N= 3–5.

**Figure 6 cells-09-01292-f006:**
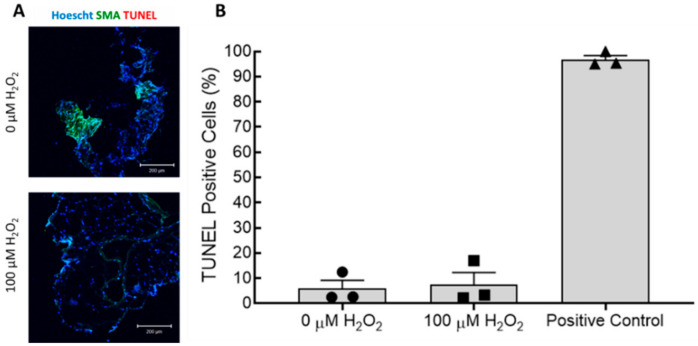
TUNEL staining confirmed that seven-day treatment of TEBVs with 100 µM H_2_O_2_ did not cause apoptosis (As in Figure 4, treatment began after maturation of TEBVs for seven days under flow). (**A**) TEBV cross-sections were stained for α-SMA and DNA strand breaks (TUNEL stain). There was no observable difference in the degree of apoptosis between TEBVs treated with 100 µM H_2_O_2_ and controls. Scale bars represent 200 µm. (**B**) Quantification of immunofluorescence showed that there was no significant increase in the number of TUNEL positive (apoptotic) cells in TEBVs treated with 100 µM H_2_O_2_. Bars represent mean ± SEM.

**Figure 7 cells-09-01292-f007:**
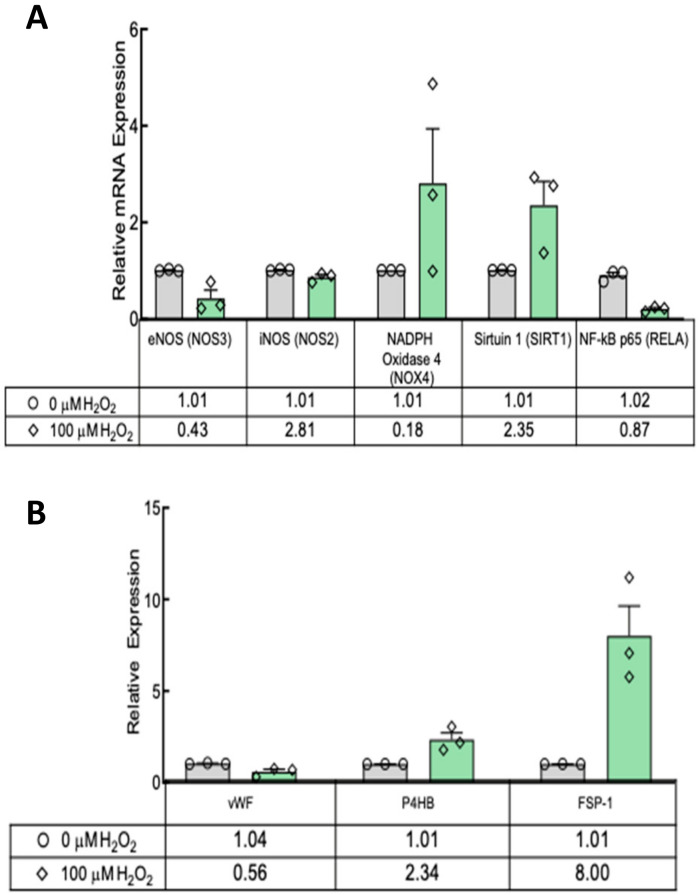
qRT-PCR of TEBVs shows SIRT1 expression increased significantly in TEBVs treated with 100 µM H_2_O_2,_ and there was a reduction in CBECFC-specific protein expression. (**A**) eNOS levels decreased significantly, but iNOS expression increased nearly threefold in response. NOX4 expression also decreased significantly with H_2_O_2_ treatment. NF-ĸB expression was unaffected. (**B**) Expression of vWF specific to the CBECFCs was decreased in TEBVs treated with 100 μM H_2_O_2_ compared to controls. The relative expressions of hNDF-specific proteins, P4HB and FSP-1, were increased in treated vessels compared to controls. N = 3. Bars represent the mean ± SEM. # indicates *p* < 0.01. * indicates *p* < 0.05.

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
