# Peer review of "Application of Oxidative Stress to a Tissue-Engineered Vascular Aging Model Induces Endothelial Cell Senescence and Activation"

_cells, 2020, doi:10.3390/cells9051292_

Round 1

Reviewer 1 Report

In the present work authors develop an in vitro TEBV model for vascular senescence and examine the relative influence of endothelial cell and smooth muscle cell senescence on vasoreactivity.

The main idea of this manuscript is interesting and the present topic appears timely.

The work is complex and the results are attractive, consistent and fair.

The manuscript is general well assembled, organized and wrote. The data are well and clear presented.

This paper could be published only after the correction of a few drawbacks, as follow:

  1. The ‘Materials and Methods’ section, would benefit of a suggestive schematic representation of in vitro model of TEBV.
  2. At ‘Results’ section the authors should specify the number of used samples, the significant statistical differences (p) and also to indicate numerical values of their results or the percentages of increase or decrease.
  3. Some reservations are about the evaluation of the TVEB function, respectively contraction and relaxation.

The authors evaluate the TVEB function by changes of diameter in the presence of agonists, vasoconstrictors or vasodilators. But, this is monitored under a microscope.

How can authors distinguish under the microscope those slight variations of diameter? Usually the function of the blood vessels is investigated using the technique of the myograph or other similar devices.

If the authors do not have this possibility, maybe they can better explain how they notice these increases or decreases in the diameter of TEBV under a microscope, what program they use for quantifications, etc.

Author Response

See attached please. 

Reviewer 2 Report

Salmon et al. aimed to evaluate endothelial cell senescence in a model of tissue engineered blood vessels, which recapitulates in vitro the physiopathology of arteries. Oxidative stress (application of H2O2) induced endothelial cell senescence, as assessed by increased p21 levels and reduces NOS3 expression. The topic is timely and relevant. The manuscript is well-written and clear. Conclusion are supported by the results. I have only minor comments:

Page 1 – line 30 typo: “As we age, vascular cells accumulate damage in a variety of ways”

The exposure to 0 H2O2 had an effect on the expression of P21 after 4 and 7 days, even if not significant. This is the effect of the physiological ageing of cells? Please discuss.If this is considered as baseline, it could be better to modify figure 1 expressing the fold change increase in P21 expression at 50 and 100 H2O2 concentration, as ratio with the baseline. The same applies also to figure 2, 3, 4, 5, and 6.

Page 6 – lines 252-282: If I understood well, the authors observed an effect on endothelium vasoreactivity with a 5-7 days treatment after an initial 7-days maturation, but not in cells directly evaluated at 5 or 7 days. For this reason, cells were evaluated at 12 and 14 days (see line 260: “…from Day 7-12 or Day 7-14 there is a corresponding increase…”). This is not well explained in the text and also in Figure 4 (panels C and D). Please clarify.

Page 8 – lines 308-309: “cross-sectional TUNEL staining confirms that treatment with 100 μM H2O2 did not affect the proportion of apoptotic cells in the TEBVs (Figure 6)”. The assay was performed after 5 or 7 days H2O2-treatment?

Replace all column histograms with scatter/dot plots. I believe that in this way graphs would be better informative, and distribution of data will be easily visualized.

Please highlight elements of novelty, strengths and limitations of the present study in the discussion section.
